# E-DDPG: Dual-Objective Deep Deterministic Policy Gradient for MRI Acceleration and Disease Classification

## Abstract

Long acquisition times remain a major challenge in clinical MRI, where a fundamental trade-off exists between the acceleration achieved through undersampling and the diagnostic utility of the reconstructed images. We cast the problem of acquiring MRI data within a fixed time budget as a discrete reinforcement learning (RL) task and propose an algorithm based on Deep Deterministic Policy Gradient, referred to as E-DDPG. E-DDPG jointly optimizes sampling patterns, image reconstruction quality, and diagnostic accuracy. We introduces three key innovations: (1) a composite reward function that simultaneously encourages cross-entropy reduction, structural similarity improvement, and decrease in predictive entropy; (2) a percentile-based replay buffer that diversifies learning by equally sampling low- and high-value transitions; and (3) integration of the Straight-Through Gumbel-Softmax mechanism to preserve end-to-end differentiability while enabling discrete action selection. We evaluate E-DDPG against state-of-the-art RL-based methods and ablation variants on the fastMRI/fastMRI+ knee datasets at acceleration factors of 4X, 8X, and 10X, demonstrating its superior performance and validating the effectiveness of each proposed component.

## 1 Introduction

With its exceptional soft tissue contrast, magnetic resonance imaging (MRI) has been the method of choice for diagnosing various cancers (Stabile et al., 2020), neurological disorders (Elmore et al., 1998), and musculoskeletal conditions (Dean Deyle, 2011). However, compared to other diagnostic imaging modalities, MRI typically requires a long acquisition time (Brown et al., 2014; Bernstein et al., 2004), which poses significant clinical challenges. This limitation has drawn considerable attention from the research community, leading to advances in rapid/fast imaging techniques such as parallel imaging (Pruessmann et al., 1999; Griswold et al., 2002), compressed sensing (Lustig et al., 2007), and deep learning-based MR image reconstruction (Hammernik et al., 2018).

The common underlying strategy among these accelerated MRI approaches is acquiring fewer raw data in Fourier space ($k$-space in MRI), often below the Nyquist sampling criterion, and reconstructing "diagnostic quality" images through sophisticated algorithms instead of straightforwardly applying the inverse Fourier transform. Consequently, as the number of acquired samples decreases (or equivalently, as higher acceleration is achieved), it becomes increasingly critical to select the most diagnostically informative samples, since highly accelerated imaging inevitably leads to image quality degradation. A promising approach may involve an intelligent system that adaptively selects $k$-space samples *to maximize predefined diagnostic criteria for a given time budget, thereby balancing acquisition efficiency with diagnostic accuracy.* While traditional optimization and supervised learning (SL) methods have demonstrated success in improving $k$-space sampling strategies and diagnostic performance, these approaches typically rely on fixed objectives—often model-based in traditional optimization or driven primarily by training data in SL (Zeng et al., 2021).

Reinforcement learning (RL) approaches have recently shown promise as compelling alternatives to traditional model-based optimization and SL methods, offering flexible and adaptive solutions (Du et al., 2024; Yang & Dong, 2024; Yen et al., 2024; Liu et al., 2024). Notable studies include ASMR (Yen et al., 2024) which learns an adaptive policy (via proximal policy optimization or

PPO) to select $k$-space samples for direct pathology classification; PG-MRI (Bakker et al., 2020), which sequentially selects $k$-space measurements via a greedy policy-gradient search; Pineda et al. (Pineda et al., 2020) which employs deep Q-learning (DQL) (Mnih et al., 2015) for sequential $k$-space column selection guided by reconstruction quality; and Xu and Oksuz (Xu & Oksuz, 2025), utilizing a PPO-based (Schulman et al., 2017) sampler that emphasizes lesion-specific fidelity.

While these existing RL approaches have demonstrated promising results, opportunities remain to more comprehensively address objectives such as diagnostic accuracy, image reconstruction quality, and predictive uncertainty. To this end, we build upon the deep deterministic policy gradient (DDPG) algorithm (Lillicrap et al., 2015), leveraging its demonstrated capability in complex, high-dimensional decision-making scenarios (Sumiea et al., 2024). Our goal is to address key limitations inherent in directly applying standard DDPG to the joint optimization of MRI acceleration and disease diagnosis. Specifically, we propose:

1. a novel SL-based *composite reward function* that jointly accounts for diagnostic accuracy, reconstruction quality, and predictive uncertainty;

2. a *percentile-based replay buffer* that partitions transitions into high- and low-reward groups, enabling balanced batch sampling to mitigate overestimation bias and improve training stability;

3. the use of differentiable discrete action selection via the *Straight-Through Gumbel-Softmax estimator* (Bengio et al., 2013; Jang et al., 2016), which enhances exploration and improves gradient flow during learning.

Finally, we evaluate the proposed algorithm, referred to as *Enhanced DDPG (E-DDPG)*, on the fastMRI (Zbontar et al., 2018) and fastMRI+ (Zhao et al., 2022) knee datasets, and compare its performance against competing RL-based methods at acceleration factors of 4X, 8X, and 10X.

## 2 THEORY

### 2.1 PROBLEM DESCRIPTION

Let $N_r$ and $N_p$ denote the numbers of frequency-encoding (readout) and phase-encoding lines, respectively. Let $C$ denote the numbers of coil elements, and let $\mathbf{x}_1, \ldots, \mathbf{x}_C \in \mathbb{C}^{N_r \times N_p}$ represent the multi-coil MRI raw data matrices. In Cartesian MRI, data acquisition involves playing out different phase-encoding gradients prior to readout, which corresponds to sequentially filling the columns of the MRI raw data matrices $\mathbf{x}_1, \ldots, \mathbf{x}_C$ (Brown et al., 2014). Then, the multi-coil MR images $\mathbf{u}_1, \ldots, \mathbf{u}_C \in \mathbb{C}^{N_r \times N_p}$ are reconstructed via the inverse Fourier transform, i.e., $\mathbf{u}_c = \mathcal{F}^{-1}(\mathbf{x}_c)$ for $c = 1, \ldots, C$. A common method for combining the individual coil images is the root-sum-of-squares (RSS) reconstruction, defined elementwise as $\mathbf{u}_{\text{RSS}}(i, j) = (|\mathbf{u}_1(i, j)|^2 + \cdots + |\mathbf{u}_C(i, j)|^2)^{1/2}$ for $1 \leq i \leq N_r$ and $1 \leq j \leq N_p$. Downstream disease diagnosis and annotation are performed by radiologists reading/interpreting the reconstructed image $\mathbf{u}_{\text{RSS}}$, which in the case of binary classification can be expressed as a mapping $\mathbf{u}_{\text{RSS}} \mapsto \mathcal{L}(\mathbf{u}_{\text{RSS}}) \in \{0, 1\}$, where 0 and 1 indicate the absence and presence of the suspected disease or condition, respectively.

Let $K \leq N_p$ be the sampling budget (i.e., the number of phase-encoding lines/columns allowed to be acquired), and let $m_K \in \{0, 1\}^{N_p}$ be an $N_p$-dimensional column vector with $\sum_{j=1}^{N_p} m_K(j) = K$. The corresponding $k$-space sampling mask is then given by $\mathbf{m}_K = \mathbf{1}_{N_r} m_K^\top \in \{0, 1\}^{N_r \times N_p}$, where $\mathbf{1}_{N_r}$ is a column vector of ones of length $N_r$ (readout length), and $m_K^\top$ is a row-vector indicating which phase-encoding columns are selected. Let $\mathcal{R}_\nu$ denote a neural network (NN) parameterized by $\nu$ trained for multi-coil MRI reconstruction. The NN-based reconstruction is then written as $\mathbf{y}_{\text{RSS}} = \mathcal{R}_\nu(\mathbf{m}_K \odot \mathbf{x}_1, \ldots, \mathbf{m}_K \odot \mathbf{x}_C)$, where $\odot$ denotes the element-wise product. Let $\mathcal{L}_\xi$ denote a NN-based binary classifier parameterized by $\xi$. The absence and presence of the suspected disease, e.g., meniscus tears, can be mapped via $z = \mathcal{L}_\xi(\mathbf{y}_{\text{RSS}}) \in \{0, 1\}$.

The goal of this paper is to determine an optimal sampling mask $\mathbf{m}_K^*$ from the possible candidates such that it maximizes the quality of the reconstructed image $\mathbf{y}_{\text{RSS}}$ and the accuracy of the predicted label $z$ via a reinforcement learning (RL) framework. To this end, we first model the problem as a Markov decision process (MDP).

## 2.2 1-STEP/STATE MDP

**State & Action Spaces** We aim to design an RL agent that selects $K$ columns *simultaneously*. Let $\mathbf{m}_{\text{init}} = \mathbf{1}_{N_r} m_{\text{init}}^\top$ denote the initial binary sampling mask, where $m_{\text{init}} \in \{0,1\}^{N_p}$ has ones at the predefined central $k$-space column indices and zeros elsewhere. The underlying MDP reduces to a *one-step/state* MDP in which the state space $\mathcal{S}$ is a singleton set, i.e., $\mathcal{S} = \{\mathbf{s}_{\text{init}}\}$ with

$$\mathbf{s}_{\text{init}} = \mathcal{R}_\nu(\mathbf{m}_{\text{init}} \odot \mathbf{x}_1, \ldots, \mathbf{m}_{\text{init}} \odot \mathbf{x}_C) \in \mathbb{R}^{N_r \times N_p}. \tag{1}$$

The action corresponds to selecting a full set of $K$ phase-encoding columns, including the predefined central $k$-space lines. Formally, the action space is defined as

$$\mathcal{A} = \left\{ m_K \in \{0,1\}^{N_p} : ||m_K||_1 = K \text{ and } m_K(j) = 1 \, \forall j \in \text{supp}(m_{\text{init}}) \right\}, \tag{2}$$

where $\text{supp}(m_{\text{init}}) \subseteq \{1, \ldots, N_p\}$ denotes the set of indices corresponding to the predefined central lines. Let $a_K^{(\ell)}$ denote a specific action $m_K^{(\ell)} \in \mathcal{A}$, where $\ell = 1, \ldots, |\mathcal{A}|$ indexes the total number of valid candidates $|\mathcal{A}| = \binom{N_p - ||m_{\text{init}}||_1}{K - ||m_{\text{init}}||_1}$. The corresponding sampling mask constructed from this action is then given by $\mathbf{m}_K^{(\ell)} = \mathbf{1}_{N_r}(a_K^{(\ell)})^\top \in \{0,1\}^{N_r \times N_p}$.

Suppose that environment transitions are deterministic. Upon selecting an action $a_K$, the terminal state is given by

$$\mathbf{s}_{\text{term}} = \mathcal{R}_\nu(\mathbf{m}_K \odot \mathbf{x}_1, \ldots, \mathbf{m}_K \odot \mathbf{x}_C) \in \mathbb{R}^{N_r \times N_p}. \tag{3}$$

For both reconstructions $\mathbf{s}_{\text{init}}$ and $\mathbf{s}_{\text{term}}$, we use the first stage of the *PromptMR* model (Xin et al., 2023) for $\mathcal{R}_\nu$. This model completes the missing $k$-space columns in $\mathbf{m}_{\text{init}} \odot \mathbf{x}_c$ and $\mathbf{m}_K \odot \mathbf{x}_c$ for $c = 1, \ldots C$, applies the inverse Fourier transform $\mathcal{F}^{-1}$ to reconstruct the individual coil images, and combines them using the RSS reconstruction.

**Reward Function** Recall that $\mathcal{L}_\xi$ is a NN-based binary classifier. We adapt the standard ResNet-50 architecture (He et al., 2016), pretrained on ImageNet (Deng et al., 2009), for $\mathcal{L}_\xi$ by: 1) inserting spatial dropout layers after each ReLU activation to improve generalization, and 2) replacing the final fully connected layer with a three-layer multilayer perceptron (MLP) that outputs two logits for binary classification. These logits are passed through a softmax function to obtain a predictive probability distributions. Let $p_{\text{init}} = \mathcal{L}_\xi(\mathbf{s}_{\text{init}})$ and $p_{\text{term}} = \mathcal{L}_\xi(\mathbf{s}_{\text{term}})$ denote the softmax output vectors at the initial and terminal states. Let $z_{\text{true}} \in \{0,1\}$ denote the ground truth label. We define the following *composite reward:*

$$r = \lambda_1(\text{CE}(p_{\text{init}}, z_{\text{true}}) - \text{CE}(p_{\text{term}}, z_{\text{true}})) + \lambda_2 \, \text{SSIM}(\mathbf{s}_{\text{init}}, \mathbf{s}_{\text{term}}) + \lambda_3(\text{H}(p_{\text{init}}) - \text{H}(p_{\text{term}})). \tag{4}$$

Here, $\text{CE}(\cdot, \cdot)$ denotes a weighted binary cross-entropy loss, where class weights are derived from the label distribution; $\text{H}(\cdot)$ is the Shannon entropy of the model's predictive distribution. The first term rewards reductions in cross-entropy (i.e., improvements in classification accuracy), the second term rewards increased structural similarity (SSIM) between the initial and terminal reconstructions, and the third term rewards decreased output entropy, encouraging the agent to make more confident predictions. Finally, $\lambda_1$, $\lambda_2$, and $\lambda_3$ are tuning parameters.

## 2.3 RL AGENT

**DDPG for 1-Step/State MDP** We apply the DDPG method (Lillicrap et al., 2015) to our one-step MDP. Let $\mathcal{Q}_\phi : \mathcal{S} \times \{0,1\}^{N_p} \to \mathbb{R}$ denote the critic network parameterized by $\phi$, and let $\mathcal{D}$ denote a replay buffer, which is a set of transitions/experiences $(\mathbf{s}_{\text{init}}, a_K, r)$. The temporal difference (TD) loss for training the critic network simplifies to

$$L(\phi) = \mathbb{E}_{(\mathbf{s}_{\text{init}}, a_K, r) \sim \mathcal{D}}[(r - \mathcal{Q}_\phi(\mathbf{s}_{\text{init}}, a_K))^2]. \tag{5}$$

Since the environment terminates immediately after a single action, the target $\mathcal{Q}$-value does not include any bootstrapped estimate from a subsequent step. Therefore, the reward $r$ itself serves as the target value for training the critic.

Let $g_1, \ldots, g_{N_p}$ be independent and identically distributed samples from $\text{Gumbel}(0,1)$. Then, the deterministic actor $\mu_\theta : \mathcal{S} \to \mathbb{R}^{N_p}$ (logits), parameterized by $\theta$, is trained to maximize the expected $\mathcal{Q}$-value:

$$J(\theta) = \mathbb{E}_{\mathbf{s}_{\text{init}} \sim \mathcal{D}}[\mathcal{Q}_\phi(\mathbf{s}_{\text{init}}, \text{top-K}(\mu_\theta(\mathbf{s}_{\text{init}}) + \mathbf{g}))], \tag{6}$$

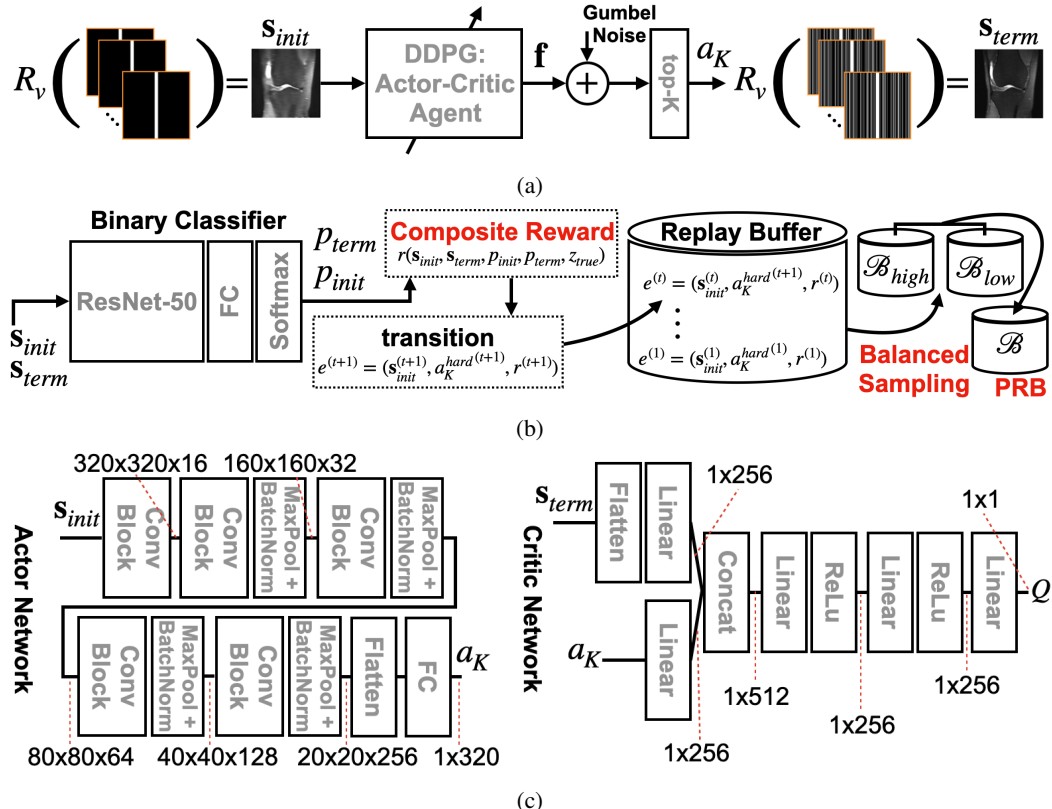

Figure 1: Pipeline of the proposed method. The forward pass of E-DDPG is illustrated in (a); data flow for the proposed composite reward, percentile-based replay buffer (PRB), and balanced sampling for batch creation are illustrated in (b). The implemented actor and critic network architectures are illustrated in (c).

where $\mathbf{g} := (g_1, \ldots, g_{N_p})$ and top-K$(\cdot)$ is a top-k operator that selects $K$ largest elements from its $N_p$-dimensional argument vector. Note that the Gumbel noise $\mathbf{g}$ is added to $\mu_\theta(\mathbf{s}_{\text{init}})$ for exploration during learning. To update the actor parameters, we employ the policy gradient method. Ideally, the gradient is given by the chain rule as follows:

$$\nabla_\theta J(\theta) = \mathbb{E}_{\mathbf{s}_{\text{init}} \sim \mathcal{D}}[\nabla_a \mathcal{Q}_\phi(\mathbf{s}_{\text{init}}, a)|_{a=\text{top-K}(\mu_\theta(\mathbf{s}_{\text{init}})+\mathbf{g})} \nabla_b \text{top-K}(b)|_{b=\mu_\theta(\mathbf{s}_{\text{init}})+\mathbf{g}} \nabla_\theta \mu_\theta(\mathbf{s})|_{\mathbf{s}=\mathbf{s}_{\text{init}}}]. \quad (7)$$

However, since the top-K operation is non-differentiable, direct backpropagation of gradients through it is obstructed. To address this, we adopt the Straight-Through Gumbel-Softmax (STGS) estimator (Bengio et al., 2013; Jang et al., 2016), which constructs a differentiable surrogate action.

**STGS for Surrogate Action** Let $\mathbf{f} := (f_1, \ldots, f_{N_p}) \in \mathbb{R}^{N_p}$ denote the logits output by the actor network, i.e., $\mathbf{f} = \mu_\theta(\mathbf{s}_{\text{init}})$. During the forward propagation, we add Gumbel noise $\mathbf{g}$ to the logits output followed by the top-K operation to select discrete actions for interaction with the environment, i.e., $a_K = \text{top-K}(\mathbf{f} + \mathbf{g}) =: a_K^{\text{hard}}$. To enable differentiability and gradient propagation during training, we employ the STGS surrogate. Specifically, we approximate/relax the top-K selection $a_K^{\text{hard}}$ as $a_K^{\text{soft}}(j) = \exp((f_j + g_j)/\tau) / \sum_{j=1}^{J} \exp((f_j + g_j)/\tau) \in [0, 1]$, where $\tau > 0$ is the temperature parameter. Then, we construct the STGS surrogate action $a_K^{\text{STGS}}$ by combining $a_K^{\text{hard}}$ and $a_K^{\text{soft}}$ in a way that ensures discrete forward behavior but maintains differentiability for gradient flow:

$$a_K^{\text{STGS}} = a_K^{\text{hard}} + (a_K^{\text{soft}} - (\text{detach}(a_K^{\text{soft}}))), \quad (8)$$

where $\text{detach}(\cdot)$ denotes the stop-gradient operator, which blocks gradients from flowing through its argument. This construction allows the surrogate action to reduce to $a_K^{\text{STGS}} = a_K^{\text{hard}} = a_K$ during the forward pass, while gradients are backpropagated through $a_K^{\text{soft}}$ only, because $a_K^{\text{hard}}$ and $\text{detach}(a_K^{\text{soft}})$ have no gradient contribution, i.e., $\partial a_K^{\text{STGS}}/\partial \mathbf{f} = \partial a_K^{\text{soft}}/\partial \mathbf{f}$.

**Percentile-based Replay Buffer (PRB)**   Intuitively, certain transitions and/or experiences—i.e., $(\mathbf{s}_{\text{init}}, a_K, r)$—that improve diagnostic confidence or significantly enhance image reconstruction are more valuable for training. Uniform sampling from the replay buffer $\mathcal{D}$ of size $N_{\mathcal{D}}$, $\mathcal{D} = \{(\mathbf{s}_{\text{init}}^{(1)}, a_K^{(1)}, r^{(1)}), \ldots, (\mathbf{s}_{\text{init}}^{(N_{\mathcal{D}})}, a_K^{(N_{\mathcal{D}})}, r^{(N_{\mathcal{D}})})\}$, may overlook these *critical transitions.* To address this, we propose a percentile-based replay buffer (PRB) that employs a dynamic prioritization mechanism mitigating the limitations of uniform sampling by: 1) separating high-reward and low-reward transitions, and 2) enforcing balanced training via a configurable sampling strategy.

The PRB divides transitions from the buffer $\mathcal{D}$ into two disjoint subsets: a high-reward buffer $\mathcal{B}_{\text{high}}$ and a low-reward buffer $\mathcal{B}_{\text{low}}$ such that $\mathcal{D} = \mathcal{B}_{\text{high}} \cup \mathcal{B}_{\text{low}}$. Transitions with rewards above a dynamically computed threshold are stored in $\mathcal{B}_{\text{high}}$, while those with rewards below the threshold are stored in $\mathcal{B}_{\text{low}}$. To be specific, for the set of rewards stored within a sliding window of recent transitions, denoted as $\mathcal{W}$, we define the threshold $\eta$ as $P$-th percentile of the reward distribution i.e., $\eta = \text{Percentile}(\mathcal{W}, P)$. Using this threshold, transitions are partitioned into a high-reward buffer $\mathcal{B}_{\text{high}}$ and a low-reward buffer $\mathcal{B}_{\text{low}}$ such that $(\mathbf{s}_{\text{init}}, a_K, r) \in \mathcal{B}_{\text{high}}$ if $r > \eta$, otherwise $(\mathbf{s}_{\text{init}}, a_K, r) \in \mathcal{B}_{\text{low}}$. This partitioning ensures that transitions associated with relatively high rewards—such as those improving diagnostic confidence or reconstruction quality—are emphasized during training, while still retaining lower-reward transitions to preserve stability and policy diversity.

For balanced learning, transitions are sampled from both $\mathcal{B}_{\text{high}}$ and $\mathcal{B}_{\text{low}}$ according to a configurable mix ratio $\beta \in [0, 1]$. Specifically, a training batch $\mathcal{B}$ of size $N_{\mathcal{B}}$ is constructed as $\mathcal{B} = S_{\mathcal{B}_{\text{high}}}(\lfloor N_{\mathcal{B}} \cdot \beta \rfloor) \cup S_{\mathcal{B}_{\text{low}}}(N_{\mathcal{B}} - \lfloor N \cdot \beta \rfloor)$, where $S_{\mathcal{C}}(q)$ is a uniform sampling operator that draws $q$ transitions from buffer $\mathcal{C}$. By maintaining this hierarchical process, the buffer dynamically adapts to prioritize transitions that are more relevant to the agent's current policy, while exploring a broad range of experiences. Finally, the complete framework is illustrated in Figure 1.

# 3   Experimental Methods

Experiments were carried out using multi-coil $k$-space data from the fastMRI knee dataset (Zbontar et al., 2018), supplemented with slice-level annotations provided by the fastMRI+ dataset (Zhao et al., 2022). Our objective was to reconstruct high-quality images from 4X, 8X, and 10X accelerated (i.e., partially sampled) $k$-space data while preserving the diagnostic information required for accurate identification of meniscus tears.

**Data Preparation & Preprocessing**   The inverse Fourier transform $\mathcal{F}^{-1}$ was applied to the fully sampled multi-coil $k$-space data. Since original image dimensions varied, each image was cropped to $320 \times 320 \times C$ (Xu & Oksuz, 2025). These cropped images were subsequently Fourier transformed back to $k$-space, resulting in a dataset with dimensions identical to the cropped images except for the coil dimension. An initial sampling mask (consisting of 16 central $k$-space columns set to 1s) was generated for each slice and coil.

**Training Strategy**   Overall, our E-DDPG framework consists of four learnable modules: 1) reconstruction network $\mathcal{R}_\nu$, i.e., PromptMR (Xin et al., 2023), 2) classification network $\mathcal{L}_\xi$, i.e., customized ResNet-50 (He et al., 2016), 3) critic network $\mathcal{Q}_\phi$, and 4) actor network $\mu_\theta$. The reconstruction network $\mathcal{R}_\nu$ and the classifier $\mathcal{L}_\xi$ were pretrained and remained frozen during subsequent training of the RL agent ($\mathcal{Q}_\phi$ and $\mu_\theta$). This *decoupling* prevents instability that might arise from simultaneously adapting multiple sets of neural network parameters for reconstruction, classification, critic, and actor losses.

The released version[1] of PromptMR was trained without modification on fully-sampled as well as 4X, 8X, and 10X accelerated $k$-space data (19,912 slices in total) for 50 epochs. All optimizer settings and architectural hyperparameters (e.g., the number of prompt blocks and channel counts) are listed in Appendix. The resultant weights $\theta$ were kept fixed for all subsequent stages.

A standard ResNet-50 architecture, pretrained on ImageNet, was adapted by 1) inserting spatial dropout layers after each ReLU activation to enhance generalization, and 2) replacing the final fully connected layer with a three-layer MLP that outputs two logits for binary classification. To address

---

[1]https://github.com/hellopipu/PromptMR

---

**Algorithm 1** Enhanced DDPG (E-DDPG) Framework

---

**Initalization:** Replay buffers $\mathcal{B}_{\text{high}}$, $\mathcal{B}_{\text{low}}$, hyperparameters: $\lambda_1, \lambda_2, \lambda_3$, percentile $P$, mixing ratio $\beta$, Gumbel temperature $\tau$, replay buffer size $N_{\mathcal{D}}$, batch size $N_{\mathcal{B}}$.

**At each batch iteration**:

1. Receive current state $\mathbf{s}_{\text{init}}$
2. Compute logits $\mathbf{f}$ from actor network $\mu_\theta(\mathbf{s}_{\text{init}})$ for each action $a_K$
3. Sample Gumbel variables: $\mathbf{g} \sim \text{Gumbel}(0, 1)$
4. Select discrete actions via STGS
5. Execute actions $a_K$ and observe next state $\mathbf{s}_{\text{term}}$
6. Compute composite reward $r$ (Eq. 4)
7. Sample transitions $(\mathbf{s}_{\text{init}}, a_K, r)$ from PRB
8. Update critic $\phi$ by minimizing temporal-difference loss (Eq. 5)
9. Update actor $\theta$ using policy gradient (Eq. 7)

**Output:** Final optimized actor policy $\mu_\theta$

---

class imbalance and mitigate overfitting, two strategies were employed simultaneously: 1) dropout regularization in the classification head, and 2) oversampling of the minority class during training to ensure balanced exposure across classes. The classifier was trained to minimize the binary cross-entropy loss for 50 epochs on the following dataset: A subset of slices from the complete fastMRI dataset was labeled as positive or negative for meniscus tears according to annotations provided by fastMRI+. Corresponding multi-coil $k$-space slices were extracted based on these labels, resulting in a final dataset comprising 6596 slices (20.6% positive) for training, 1247 slices (14.6% positive) for validation, and 1502 slices (20.3% positive) for testing.

In E-DDPG, the actor $\mu_\theta$ takes the image reconstructed from partially sampled $k$-space data, i.e., $\mathbf{s}_{\text{init}}$, and feeds it through an initial convolutional block ($3\times3\,\text{Conv} \rightarrow \text{InstanceNorm} \rightarrow \text{ReLu} \rightarrow \text{Dropout}$), expanding the input to a predefined base number of feature channels. The output tensor then traverses four identical downsampling stages; each stage consists of the same convolutional block followed by a $2\times2\,\text{MaxPool} \rightarrow \text{BatchNorm}$ block, which doubles the number of feature channels while halving the spatial resolution. The output feature tensor of the final stage is flattened and passed through a three-layer MLP with $\text{Leaky-ReLU}$ activations, projecting it onto a length-$J$ score vector, where $J$ is the number of phase-encoding columns. The critic evaluates a state–action pair $(\mathbf{s}_{\text{init}}, a_K)$. The image state $\mathbf{s}_{\text{init}} \in \mathbb{R}^{320\times320}$ is first flattened and then projected via a fully connected layer. The action vector $a_K \in \mathbb{R}^{N_p}$ is passed through another fully connected layer of the same width. These two outputs are concatenated and passed through two additional ReLU-activated fully connected layers. Finally, a linear neuron returns the scalar $\mathcal{Q}_\phi(\mathbf{s}_{\text{init}}, a_K)$. The actor-critic architecture is illustrated in Figure 1, and the training procedure at each batch iteration is summarized in Algorithm 1.

All methods, including our proposed approach and the compared algorithms, were trained for 30 epochs using the same random seed. We set $\beta = 0.5$ to ensure an equal mix of high- and low-reward transitions in each batch. In all experiments we weight the three reward components with the coefficients $\lambda_1 = 10$, $\lambda_2 = 100$, and $\lambda_3 = 10$, to balance the relative influence of confidence tightening versus the other two objectives. All trainings were completed using a 4-way NVIDIA H200 GPU machine.

**Comparison** We compared the proposed E-DDPG[2] with the following methods. *1) Baseline:* The RSS coil combination of the inverse Fourier transform of the fully sampled multi-coil $k$-space data served as the reference for evaluating SSIM and PSNR. The same binary classifier used in our E-DDPG training was also trained on these RSS images, along with diagnostic annotations indicating the presence or absence of meniscus tears. This supervised learning approach served as the baseline framework for evaluating classification performance. *2) Competing RL-based MRI Acceleration:* ASMR[3] (Yen et al., 2024), Pineda et al.[4] (Pineda et al., 2020), and Xu and Oksuz[5]

---

[2]https://anonymous.4open.science/r/eddpg-8B30

[3]https://github.com/robinyen/asmr

[4]https://github.com/facebookresearch/active-mri-acquisition

[5]https://github.com/Ruru-Xu/KSRO

Table 1: Performance comparison across metrics at different acceleration factors.

| 4X | | | | |
|---|---|---|---|---|
| **Algorithm** | SSIM | PSNR | AUC | Bal. Acc. |
| Baseline | 1.000 | Inf | 0.883±0.001 | 0.782±0.001 |
| E-DDPG | **0.864±3e-3** | **32.236±1.731** | **0.857±0.004** | **0.750±0.007** |
| ASMR | 0.853±4e-3 | 31.035±2.436 | 0.834±0.006 | 0.741±0.012 |
| Pineda et al. | 0.797±5e-2 | 27.911±2.183 | 0.814±0.011 | 0.674±0.013 |
| Xu and Oksuz | 0.850±5e-3 | 30.409±2.213 | 0.831±0.005 | 0.730±0.007 |
| **Ablation Study** | SSIM | PSNR | AUC | Bal. Acc. |
| w/o STGS | 0.853±7e-3 | 30.799±2.125 | 0.828±0.003 | 0.728±0.008 |
| w/o PRB | 0.859±2e-3 | 31.092±2.006 | 0.854±0.004 | 0.740±0.007 |
| w/o R | 0.849±5e-3 | 29.704±1.445 | 0.820±0.006 | 0.714±0.008 |
| DDPG | 0.809±2e-2 | 28.404±2.226 | 0.749±0.013 | 0.708±0.009 |
| 8X | | | | |
| **Algorithm** | SSIM | PSNR | AUC | Bal. Acc. |
| Baseline | 1.000 | Inf | 0.883±0.001 | 0.782±0.001 |
| E-DDPG | **0.862±3e-3** | **30.838±1.589** | **0.854±0.006** | **0.751±0.008** |
| ASMR | 0.847±5e-3 | 29.654±1.058 | 0.829±0.008 | 0.735±0.010 |
| Pineda et al. | 0.795±4e-2 | 27.187±2.110 | 0.766±0.012 | 0.665±0.018 |
| Xu and Oksuz | 0.838±6e-3 | 29.492±2.171 | 0.823±0.004 | 0.727±0.009 |
| **Ablation Study** | SSIM | PSNR | AUC | Bal. Acc. |
| w/o STGS | 0.851±6e-3 | 29.510±2.945 | 0.822±0.008 | 0.723±0.009 |
| w/o PRB | 0.857±4e-3 | 29.697±1.741 | 0.847±0.003 | 0.736±0.004 |
| w/o R | 0.844±4e-3 | 27.865±1.234 | 0.819±0.009 | 0.716±0.005 |
| DDPG | 0.772±2e-2 | 26.483±1.859 | 0.737±0.007 | 0.691±0.018 |
| 10X | | | | |
| **Algorithm** | SSIM | PSNR | AUC | Bal. Acc. |
| Baseline | 1.000 | Inf | 0.883±0.001 | 0.782±0.001 |
| E-DDPG | **0.857±4e-3** | **30.673±1.926** | **0.823±0.006** | **0.720±0.014** |
| ASMR | 0.841±3e-3 | 29.403±2.084 | 0.819±0.008 | 0.718±0.013 |
| Pineda et al. | 0.776±1e-2 | 26.394±3.665 | 0.732±0.018 | 0.662±0.014 |
| Xu and Oksuz | 0.835±6e-3 | 29.346±2.198 | 0.810±0.010 | 0.716±0.006 |
| **Ablation Study** | SSIM | PSNR | AUC | Bal. Acc. |
| w/o STGS | 0.848±6e-3 | 27.954±1.528 | 0.808±0.005 | 0.697±0.009 |
| w/o PRB | 0.854±2e-3 | 28.062±1.266 | 0.820±0.004 | 0.714±0.012 |
| w/o R | 0.841±5e-3 | 27.546±1.871 | 0.805±0.011 | 0.689±0.012 |
| DDPG | 0.769±2e-2 | 24.464±2.025 | 0.711±0.008 | 0.675±0.021 |

(Xu & Oksuz, 2025) were chosen for comparison due to their relevance to the proposed method and availability. These methods were trained on the same 4X, 8X, and 10X accelerated multi-coil $k$-space raw data from the fastMRI knee dataset. In addition, a negative weighted cross-entropy term, i.e., $r = -\text{CE}(p_{\text{term}}, z_{\text{true}})$ in Eq. 4, was added to their original reward functions to explicitly link their training to the binary classification objective.

**Ablation Study**   To systematically analyze the individual contributions of 1) the STGS estimator, 2) PRB, and 3) composite reward, we performed the following ablation experiments. *E-DDPG without STGS (w/o STGS):* The STGS estimator was replaced by a conventional $\text{softmax}/\text{top-K}$ selector. The actor still outputs a logit vector; however, the discrete action is now obtained by selecting K largest softmax probabilities rather than through the STGS surrogate. All other elements (PRB and composite reward) remained unchanged. *E-DDPG without (w/o PRB):* The PRB was replaced by a standard DDPG replay buffer, resulting in uniform sampling of stored transitions. The STGS estimator and composite reward remained unchanged. *E-DDPG without (w/o R):* The composite reward was replaced by a single negative weighted cross-entropy term, i.e., $r = -\text{CE}(p_{\text{term}}, z_{\text{true}})$. The PRB and the STGS remained unchanged. *DDPG:* The STGS estimator, PRB, and composite reward were all removed from E-DDPG. Like E-DDPG without STGS, the same $\text{softmax}/\text{top-K}$ selector was used instead. In other words, we adapted the standard DDPG by adding $\text{softmax}/\text{top-K}$ for discrete action selection.

**Image Quality Assessment**   SSIM and PSNR were computed between reconstructed images produced by the proposed and comparison methods, and the baseline reference. Diagnostic performance was assessed using the Area Under the Receiver Operating Characteristic curve (AUC) and Balanced Accuracy (Bal. Acc.).

## 4 EXPERIMENTAL RESULTS

Overall, the proposed method (E-DDPG) consistently outperformed competing methods across all tested undersampling factors in both image fidelity (SSIM and PSNR) and diagnostic accuracy (AUC

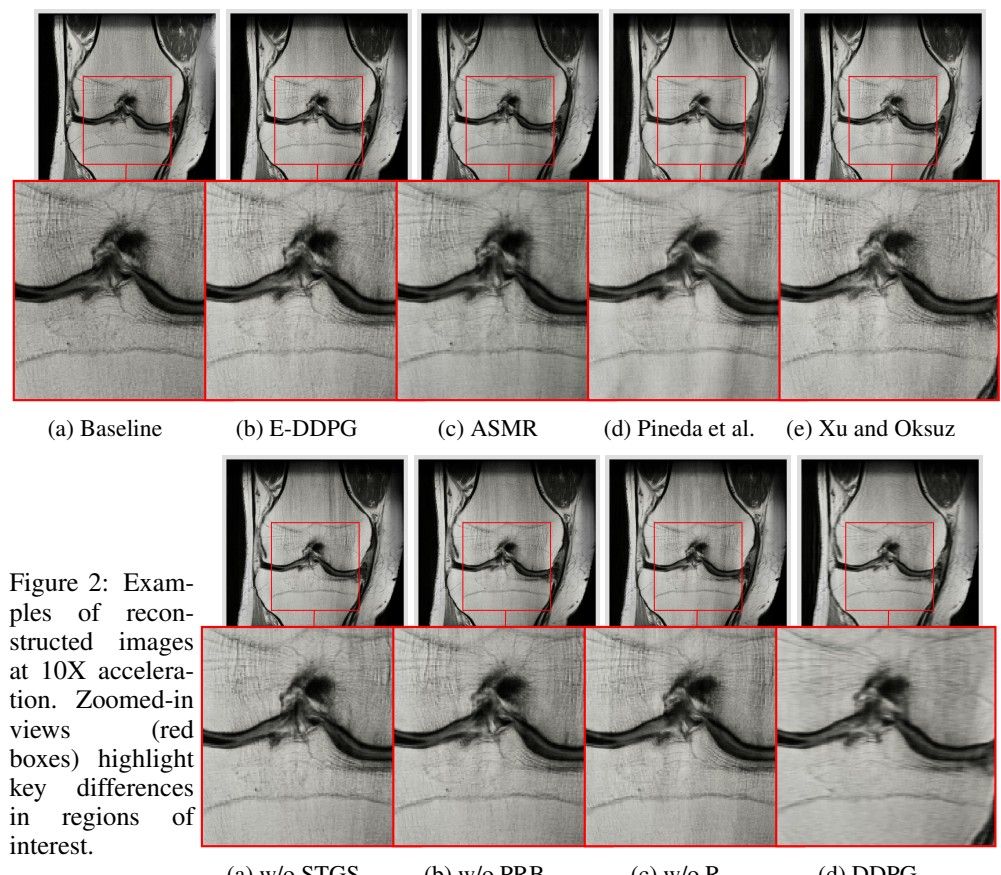

(a) Baseline      (b) E-DDPG      (c) ASMR      (d) Pineda et al.      (e) Xu and Oksuz

Figure 2: Examples of reconstructed images at 10X acceleration. Zoomed-in views (red boxes) highlight key differences in regions of interest.

(a) w/o STGS      (b) w/o PRB      (c) w/o R      (d) DDPG

and Bal. Acc.)—best results indicated in bold excluding baseline—as shown in Table 1. Across all three acceleration factors, E-DDPG achieves the strongest overall results. Its mean SSIM is $0.869 \pm 0.015$, about three points higher than the next-best XU ($0.840 \pm 0.011$) and seven to eight points better than Pineda et al. and PILOT. As for PSNR, E-DDPG records $32.87 \pm 0.12$, whereas XU reaches 30.57 and the other two algorithms stay in the 27–28 range. E-DDPG attains an average AUC of $0.847 \pm 0.020$, roughly 0.03 higher than XU and 0.07–0.08 above PILOT and Pineda et al. Its Balanced Accuracy is $0.747 \pm 0.019$, versus 0.715 for Xu and Oksuz and about 0.67 for the other two methods.

These improvements in quantitative performance translated into enhanced reconstructed image quality, as shown in Figure 2. Zoomed-in views revealed that image details obtained by E-DDPG closely match those of the baseline, with Xu and Oksuz also producing comparably clear results—consistent with quantitative findings in Table 1. In contrast, images reconstructed by the remaining methods exhibited noticeable blur, accompanied by a dark shaded region extending vertically along the right side of the images. Among the ablation variants, the DDPG exhibited the most significant image degradation. The other ablation variants (w/o STGS, w/o PRB, and w/o R) demonstrated slightly reduced image quality compared to E-DDPG, but maintained similar overall visual quality. Examples of reconstructed images and policies at 4X and 8X are in *Appendix*. Figure 4 demonstrates the mean of the metrics values and the $95\%$ confidence intervals in 4 seeds $\left( \bar{x} \pm t \frac{s}{\sqrt{n}} \right)$ (Petty, 2012).

## 5 DISCUSSION AND CONCLUSION

Within the RL + SL framework, we have proposed E-DDPG addressing three key limitations of standard DDPG. First, while vanilla DDPG is typically trained with a single scalar reward signal, potentially missing subtle but critical changes in prediction confidence and reconstruction quality, E-DDPG employs a composite reward (in Eq. 4) that integrates weighted cross-entropy reduction

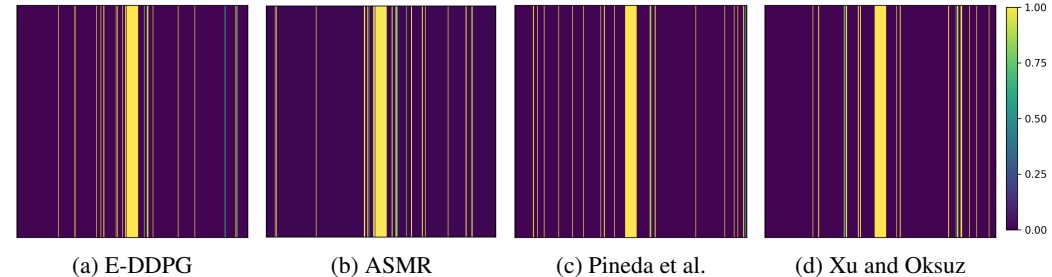

Figure 3: Normalized frequency of the sampled $k$-space columns for 10X acceleration.

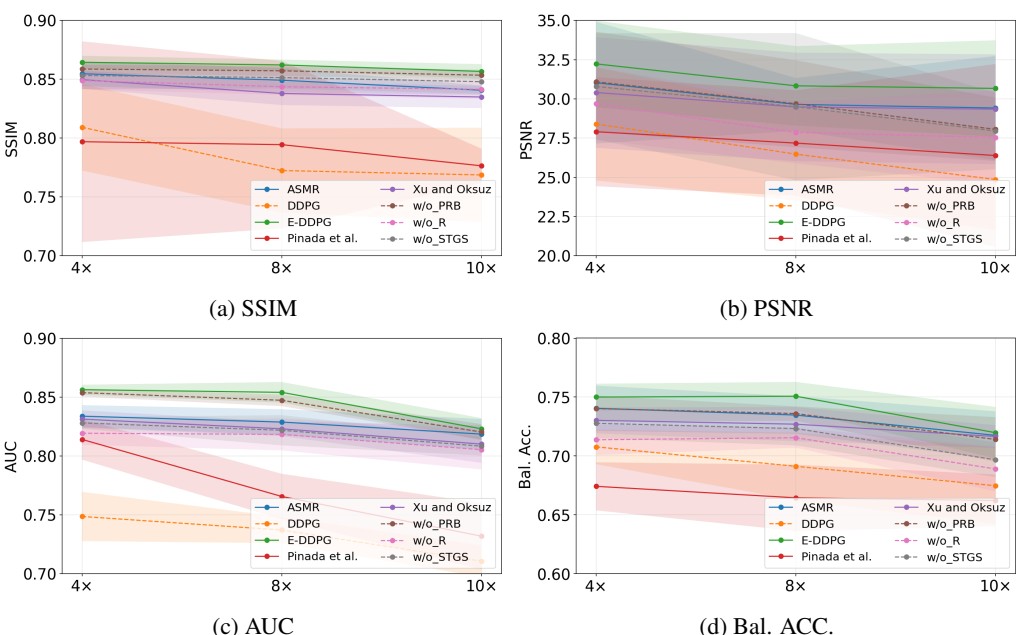

Figure 4: Confidence interval.

$\Delta$CE, SSIM improvement $\Delta$SSIM, and entropy reduction $\Delta$H. Second, vanilla DDPG uniformly samples transitions from its replay buffer, treating all experiences equally informative; E-DDPG introduces prioritized sampling so that rare or high-error experiences are replayed more frequently. Third, directly applying DDPG's continuous actor to discrete sampling is challenging because discretizing the action with a hard top-K breaks the differentiability of the actor-critic gradient chain and restricts exploration. To address this, we incorporate the STGS estimator to preserve differentiability for effective backpropagation. Learned policies, i.e., the resulting $k$-space sampling masks, at 10X acceleration are shown in Figure 8. Although it is not immediately clear which specific sampling patterns would yield the best performance, this very observation indicates the effectiveness of the proposed E-DDPG in adaptively selecting the most informative $k$-space columns.

Finally, despite the gains demonstrated by E-DDPG, several practical limitations remain. First, the framework operates on individual 2D slices. Second, the reconstruction and classifier backbones were held fixed during RL training. Third, the current approach employs an 1-step MDP formulation that selects $K$ columns simultaneously with a single action. Future work will focus on 1) 3D $k$-space trajectories coupled with a 3D reconstruction/classifier networks, 2) enabling end-to-end joint optimization for $\mathcal{R}_\nu$, $\mathcal{L}_\xi$, $\mathcal{Q}_\phi$, and $\mu_\theta$, and 3) investigation of sequential/active sampling that could offer further improvements as MRI is inherently a dynamic environment. In conclusion, we have demonstrated RL-based adaptive $k$-space sampling, achieving high-quality image reconstruction and and diagnostic accuracy at various acceleration factors.

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

# A APPENDIX

## A.1 LIST OF HYPERPARAMETERS

Table 2 lists the hyperparameters used in E-DDPG across all experiments. Table 3 lists the hyperparameters that remain fixed throughout all experiments for the two auxiliary networks used by E-DDPG. The upper half lists the architectural and training settings of the PromptMR reconstructor, while the lower half records the optimizer, learning schedule, and regularization choices for the ResNet-50 classifier.

Table 2: E-DDPG hyperparameter settings

| Parameter | Value |
|---|---|
| Optimizer | Adam |
| Learning rate | $1 \times 10^{-4}$ |
| Dropout rate | 0.25 |
| Buffer size ($N_D$) | 300000 |
| $B_{high}, B_{low}$ | 150000, 150000 |
| Batch size ($N_B$) | 128 |
| Reward coefficients ($\lambda_1, \lambda_2, \lambda_3$) | 10, 100, 10 |
| Mixing ratio ($\alpha$) | 0.5 |
| Percentile ($P$) | 0.75 |
| Gumbel temperature ($\tau$) | 1 |

Table 3: PromptMR and ResNet-50 hyperparameter settings

| Parameter | Value |
|---|---|
| **PromptMR hyperparameters** | |
| Optimizer | AdamW |
| num_cascades | 12 |
| num_adj_slices | 3 |
| n_feat0 | 48 |
| feature_dim | [72, 96, 120] |
| prompt_dim | [24, 48, 72] |
| sens_n_feat0 | 24 |
| sens_feature_dim | [36, 48, 60] |
| sens_prompt_dim | [12, 24, 36] |
| len_prompt | [5, 5, 5] |
| prompt_size | [64, 32, 16] |
| n_enc_cab | [2, 3, 3] |
| n_dec_cab | [2, 2, 3] |
| n_skip_cab | [1, 1, 1] |
| n_bottleneck_cab | 3 |
| lr | $1 \times 10^{-4}$ |
| lr_step_size | 35 epochs |
| lr_gamma | 0.1 |
| **ResNet-50 hyperparameters** | |
| Optimizer | Adam |
| Learning rate | $1 \times 10^{-4}$ |
| Scheduler step size | 8 |
| Scheduler decay factor | 0.5 |
| Dropout rate | 0.1 |

## A.2 EXPERIMENTAL RESULTS

Figures 5–8 present representative reconstructions and the $k$-space sampling patterns/statistics for 4X and 8X accelerations. Figures 5 and 7 show 4X and 8X reconstructions (with zoomed insets), while Figures 6 and 8 illustrate the normalized column-selection frequencies corresponding to these acceleration factors. The exhibited image quality in Figures 5-7 is quantitatively supported by the SSIM, PSNR, AUC, and Bal. Acc. metrics reported in the main manuscript. Figure 9 shows the mean rewards of the proposed method over 30 epochs along with different acceleration factors.

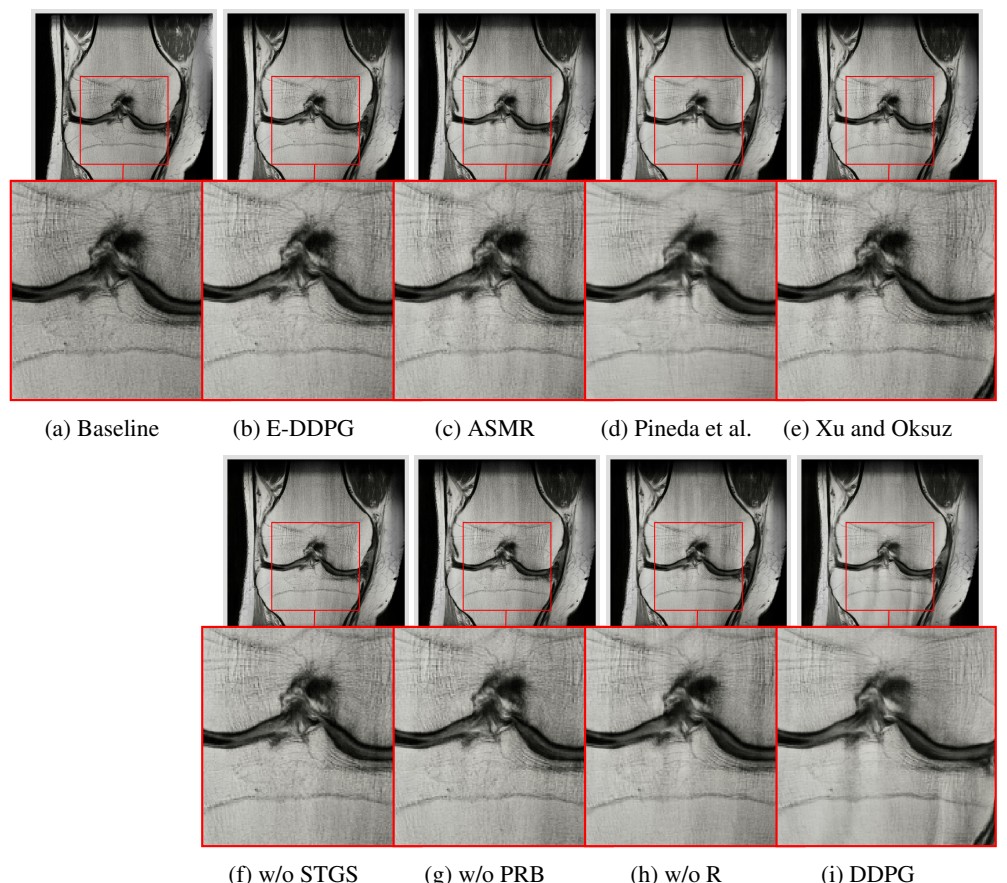

(a) Baseline   (b) E-DDPG   (c) ASMR   (d) Pineda et al.   (e) Xu and Oksuz

(f) w/o STGS   (g) w/o PRB   (h) w/o R   (i) DDPG

Figure 5: Examples of reconstructed images at 4X acceleration. Zoomed-in views (red boxes) highlight key differences in regions of interest.

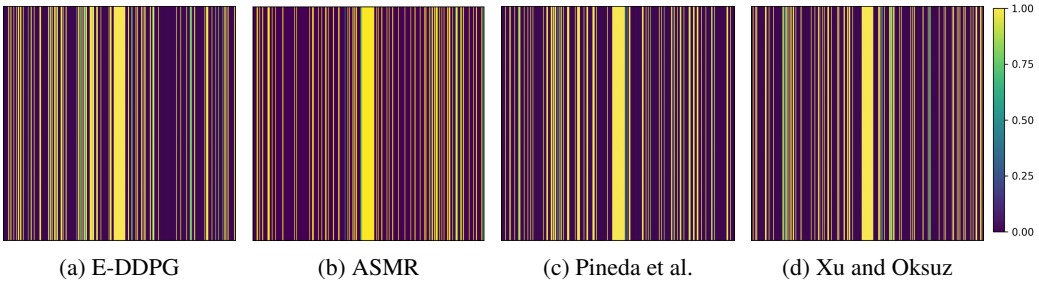

(a) E-DDPG   (b) ASMR   (c) Pineda et al.   (d) Xu and Oksuz

Figure 6: Normalized frequency of the sampled $k$-space columns for 4X acceleration.

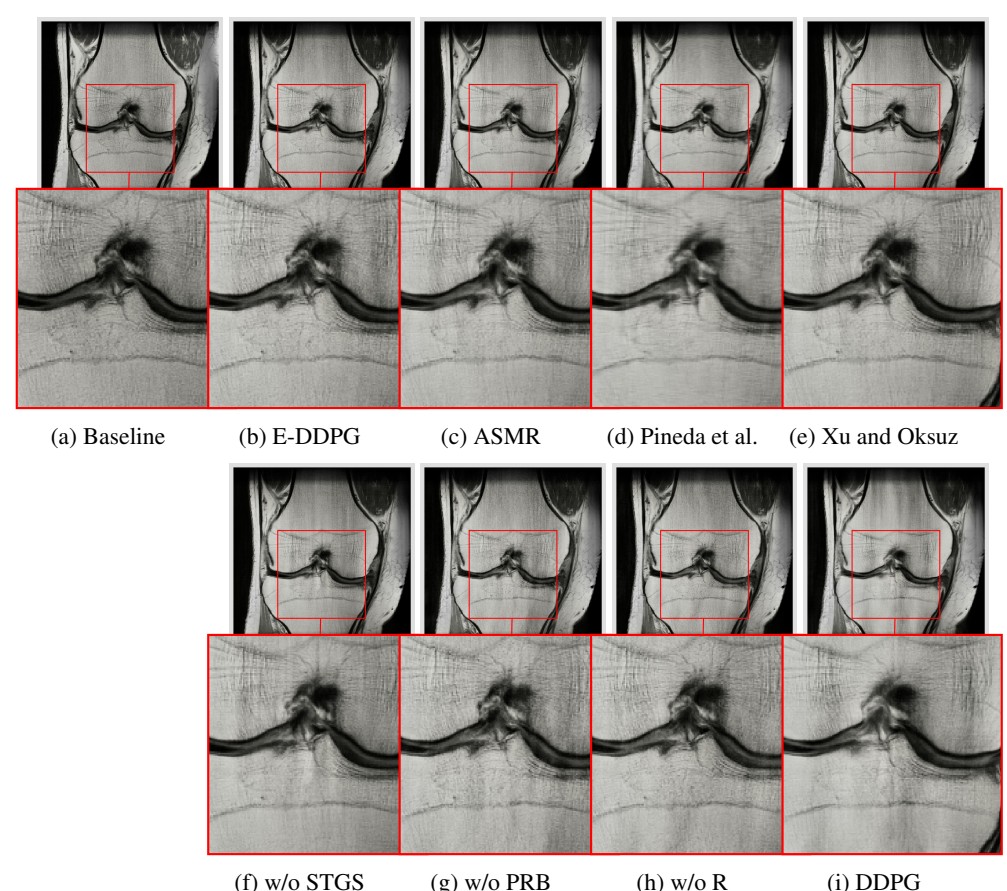

(a) Baseline     (b) E-DDPG     (c) ASMR     (d) Pineda et al.     (e) Xu and Oksuz

(f) w/o STGS     (g) w/o PRB     (h) w/o R     (i) DDPG

Figure 7: Examples of reconstructed images at 8X acceleration. Zoomed-in views (red boxes) highlight key differences in regions of interest.

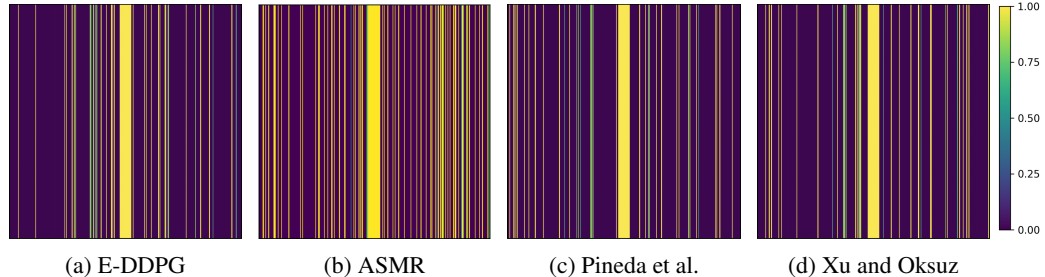

(a) E-DDPG     (b) ASMR     (c) Pineda et al.     (d) Xu and Oksuz

Figure 8: Normalized frequency of the sampled $k$-space columns for 8X acceleration.

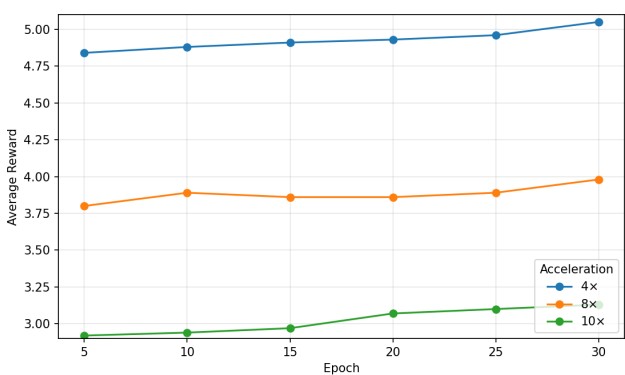

Figure 9: Convergence graph of the proposed algorithm for different acceleration factors

