# OpenReview forum: "E-DDPG: Dual-Objective Deep Deterministic Policy Gradient for MRI Acceleration and Disease Classification"
_ICLR.cc/2026/Conference — Submitted to ICLR 2026_

### Official Review · Reviewer_hnqi · 2025-11-01

**Soundness:** 3
**Presentation:** 3
**Contribution:** 3
**Rating:** 6
**Confidence:** 3

**Summary:**

This paper tackles the long-standing challenge of accelerating MRI acquisition without compromising diagnostic quality. The authors frame k-space undersampling as a discrete reinforcement learning (RL) task and propose E-DDPG (Enhanced Deep Deterministic Policy Gradient) — a dual-objective RL framework that jointly optimizes sampling efficiency, image reconstruction fidelity, and disease classification accuracy.

E-DDPG extends the traditional DDPG algorithm with three innovations:

- A composite reward function combining classification accuracy (cross-entropy reduction), reconstruction quality (SSIM), and diagnostic confidence (entropy reduction).
- A percentile-based replay buffer (PRB) that balances sampling between high- and low-reward experiences for stable training.
- A Straight-Through Gumbel-Softmax (STGS) estimator that enables differentiable discrete k-space column selection while maintaining gradient flow.

The method is evaluated on the fastMRI and fastMRI+ knee datasets under 4×, 8×, and 10× acceleration. Quantitative metrics (SSIM, PSNR, AUC, Balanced Accuracy) and qualitative results show consistent superiority of E-DDPG over PPO- and DQL-based approaches.

The paper offers engineering and application-level novelty, not theoretical innovation. However, the adaptation of DDPG with STGS and percentile replay for a dual-objective, discrete-action medical imaging task represents a meaningful and creative contribution.

**Strengths:**

- Casting MRI acquisition as a dual-objective reinforcement learning problem is conceptually compelling and addresses a clinically important trade-off: scan speed versus diagnostic accuracy.
- Each enhancement (composite reward, PRB, STGS) directly addresses specific limitations of standard DDPG — i.e., single-scalar rewards, uniform replay bias, and non-differentiable discrete actions. The proposed design is methodologically coherent and practically effective.
- Comprehensive experimental validation: benchmarks on both fastMRI and fastMRI+ datasets with 4×–10× acceleration factors; four ablation variants (removing STGS, PRB, or reward components) clearly isolate each contribution; comparisons to leading RL approaches under controlled settings ensure fairness.
- Equations are precise, architectural details are complete, and training hyperparameters are fully listed. Figures are well-labeled and readable. The authors mention public code release, which strengthens reproducibility.

**Weaknesses:**

- SSIM, PSNR, and AUC improve modestly, and the clinical impact (e.g., lesion visibility, diagnostic confidence in radiologists) is unquantified. The numerical gains, though consistent, may be marginal in practice.
- Experiments focus only on 2D knee MRIs and binary classification. No tests on other anatomies, contrasts, or 3D data are presented. It is unclear whether the learned policies generalize beyond this specific domain.
- The work omits comparisons with modern differentiable or supervised sampling optimization techniques (e.g., [1][2]). This leaves unclear whether RL offers a genuine advantage over deterministic gradient-based optimization.

References:
[1] https://arxiv.org/pdf/1901.01960
[2] https://arxiv.org/pdf/2210.12548

**Questions:**

- Why was a single-step decision process used instead of a sequential MDP that models progressive acquisition? How does this affect exploration and long-term reward learning? Would a multi-step policy (e.g., sequentially adding k-space lines) improve adaptivity or efficiency?

---

> ### Author Response · Authors · 2025-12-02
>
> > **R3.1.** The work omits comparisons with modern differentiable or supervised sampling optimization techniques (e.g., [1][2]). This leaves unclear whether RL offers a genuine advantage over deterministic gradient-based optimization. References: [1] https://arxiv.org/pdf/1901.01960 [2] https://arxiv.org/pdf/2210.12548
>
> We add comparisons with [1] and Weiss et al. (https://arxiv.org/abs/1905.09324). A publicly availabe implementation of [2] suitable for our setting could not be identified, and Weiss et al. provide a highly relevant baseline. Both methods were trained on the same fastMRI/fastMRI+ knee dataset, and the results are shown below.
>
> |Acceleration|Method|SSIM|PSNR|AUC|Bal. Acc.|
> |-|-|-|-|-|-:|
> |**4X**|**E-DDPG**|**0.864**|**31.916**|**0.855**|**0.740**|
> ||LOUPE|0.847|30.129|0.830|0.729|
> ||Weiss et al.|0.821|28.969|0.808|0.713|
> |**8X**|**E-DDPG**|**0.863**|**31.106**|**0.847**|**0.744**|
> ||LOUPE|0.840|29.918|0.826|0.728|
> ||Weiss et al.|0.812|28.568|0.786|0.709|
> |**10X**|**E-DDPG**|**0.853**|**30.840**|**0.831**|**0.717**|
> ||LOUPE|0.827|28.708|0.804|0.708|
> ||Weiss et al.|0.796|27.518|0.769|0.683|
>
> Our preliminary comparison with the suggested literature may help address the reviewer's concern, indicating that **the proposed RL approach offers an advantage over deterministic gradient-based optimization.**
>
> > **R3.2.** Why was a single-step decision process used instead of a sequential MDP that models progressive acquisition? How does this affect exploration and long-term reward learning? Would a multi-step policy (e.g., sequentially adding k-space lines) improve adaptivity or efficiency?
>
> The reviewer made a valid point. A sequential MDP is a viable alternative, and we added a small set of active sampling variants as suggested by R1 (**please see our response to R1.1**). Although the proposed 1-step formulation consistently outperforms the 4- and 10-column MDPs, drawing firm conclusions without a systematic comparison would be premature. At present, the choice is largely driven by practical considerations: **sequential/active sampling introduces per-TR latency because gradient waveforms must be computed and placed in the memory in real time, and TR is on the order of a few milliseconds.** This overhead can lengthen the scan, whereas the additional time could instead be used to acquire additional samples under a 1-step MDP.
>
> Regarding exploration and the long-term reward learning, our understanding is that this concerns whether the policy can reliably improve expected return over training in the absence of intermediate rewards. During training we inject Gumbel noise into the logits before taking the top-K to make sure that every feasible k-space location retains a non-zero selection probability and enabling broad exploration. Each sampled mask receives a terminal reward derived from reconstruction or classification quality. The critic learns to approximate the expected return for these masks, and the actor is updated to increase the likelihood of sets that yield higher predicted utility. **As training progresses and the critic becomes more accurate, the mask distribution concentrates around high-reward patterns, which is reflected in the rising reward trajectories shown in the Appendix.**

---

### Official Review · Reviewer_4ZVa · 2025-11-01

**Soundness:** 3
**Presentation:** 3
**Contribution:** 2
**Rating:** 4
**Confidence:** 3

**Summary:**

The paper proposes E-DDPG, a DDPG-based reinforcement learning framework for accelerated MRI. The goal is to jointly optimize image reconstruction and disease classification. Experiments on the fastMRI knee dataset at 4x, 8x, and 10x acceleration demonstrate the performance improvements over recent RL-based baselines.

**Strengths:**

- The paper is well-written, and the problem formulation is clear. The proposed modules are well-motivated and technically sound.

- The paper provides comparisons with existing RL-based baselines.

- The authors explicitly discuss the limitations of their approach.

- The visualization of selected k-space trajectories and the reconstructed images across methods helps understanding.

**Weaknesses:**

- It is unclear how critical the use of the PromptMR model is to overall performance. Does the proposed method rely heavily on this specific reconstruction backbone?

- The experiments are limited to knee MRI and a binary meniscus tear classification task. The work would be stronger if evaluated on more anatomies or pathologies.

- For the competing baseline solutions, a negative weighted cross-entropy term was added to the original rewards. It remains unclear how these baselines perform without this additional term.

- The sensitivity of the model performance to the reward coefficients is not analyzed.

**Questions:**

- How much does the proposed approach depend on the PromptMR backbone? Would using another reconstruction model, e.g., U-Net, VarNet.., substantially change performance?

- Could the proposed method generalize to other anatomies or pathologies?

- How sensitive is E-DDPG to the choice of reward weights?

---

> ### Author Response · Authors · 2025-12-02
>
> > **R2.1.** How much does the proposed approach depend on the PromptMR backbone? Would using another reconstruction model, e.g., U-Net, VarNet.., substantially change performance?
>
> We add a new reconstructor adapted from the official fastMRI repository's U-Net implementation (https://github.com/facebookresearch/fastMRI/tree/main/fastmri_examples/unet). All models were trained for acceleration factors of 4X, 8X, and 10X using this updated reconstructor backbone, and the results are below with the best-performing methods shown in bold.
>
> |Acceleration|Method|SSIM|PSNR|AUC|Bal. Acc.|
> |-|-|-|-|-|-:|
> |4X|**E-DDPG**|**0.821**|**29.164**|**0.829**|**0.703**|
> ||ASMR|0.807|28.154|0.810|0.691|
> ||Pineda et al.|0.779|26.014|0.728|0.680|
> ||Xu & Oksuz|0.801|27.952|0.803|0.695|
> |8X|**E-DDPG**|**0.813**|**28.771**|**0.819**|**0.696**|
> ||ASMR|0.800|27.962|0.803|0.685|
> ||Pineda et al.|0.761|25.186|0.710|0.676|
> ||Xu & Oksuz|0.794|27.259|0.775|0.672|
> |10X|**E-DDPG**|**0.801**|**28.205**|**0.800**|**0.678**|
> ||ASMR|0.791|27.305|0.774|0.668|
> ||Pineda et al.|0.750|24.108|0.701|0.664|
> ||Xu & Oksuz|0.788|26.912|0.769|0.669|
>
>
> **The proposed method consistently outperformed the competing methods when paired with the U-Net backbone.** The relative differences for each metric between PromptMR and U-Net were also computed with respect to the PromptMR values across all accelerator factors, as shown below (in [%]).
>
> |E-DDPG|SSIM|PSNR|AUC|Bal. Acc.|
> |-|-|-|-|-:|
> |4X|4.9|8.6|3.3|5.0|
> |8X|5.7|7.5|3.3|6.4|
> |10X|6.0|8.5|3.7|7.3|
>
> All relative difference were $\leq 8.5$% across all metrics and acceleration factors,indicating that **the choice of reconstruction backbone does not substantially affect the performance of the proposed method.**
>
> > **R2.2.** Could the proposed method generalize to other anatomies or pathologies?
>
> We add an additional task targeting sprain-injury detection using the proposed E-DDPG method. The training, validation, and test datasets contain 5.00%, 2.08%, and 5.59% positive labels, respectively. The results are below, with the best-performing methods shown in bold.
>
> |Acceleration|Method|SSIM|PSNR|AUC|Bal. Acc.|
> |-|-|-|-|-|-|
> |**4X**|Baseline|1.000|Inf|0.855±0.007|0.756±0.002|
> ||**E-DDPG**|**0.860±0.001**|**31.31±0.70**|**0.815±0.008**|**0.725±0.006**|
> ||ASMR|0.847±0.006|30.08±0.22|0.796±0.005|0.709±0.002|
> ||Pineda et al.|0.772±0.002|25.97±0.32|0.785±0.004|0.632±0.013|
> ||Xu & Oksuz|0.843±0.002|29.96±0.13|0.799±0.002|0.719±0.006|
> |**8X**|Baseline|1.000|Inf|0.855±0.007|0.756±0.002|
> ||**E-DDPG**|**0.857±0.002**|**30.16±0.14**|**0.803±0.002**|**0.721±0.004**|
> ||ASMR|0.833±0.006|29.18±0.67|0.784±0.005|0.701±0.004|
> ||Pineda et al.|0.760±0.001|24.97±0.18|0.762±0.002|0.605±0.004|
> ||Xu & Oksuz|0.832±0.002|29.24±0.50|0.792±0.003|0.711±0.005|
> |**10X**|Baseline|1.000|Inf|0.855±0.007|0.756±0.002|
> ||**E-DDPG**|**0.844±0.005**|**29.92±0.38**|**0.796±0.006**|**0.709±0.002**|
> ||ASMR|0.823±0.009|28.23±0.63|0.773±0.009|0.669±0.006|
> ||Pineda et al.|0.738±0.007|24.87±0.70|0.738±0.007|0.511±0.007|
> ||Xu & Oksuz|0.827±0.002|28.44±0.36|0.774±0.002|0.690±0.005|
>
> As the results suggest, **we are inclined to believe that our E-DDPG can handle other pathologies.**
>
> > **R2.3.** How sensitive is E-DDPG to the choice of reward weights?
>
> A grid search was applied by scaling each coefficient fivefold for the 4X setting on the same dataset. The results are shown below.
>
> |$(\lambda_1,\lambda_2,\lambda_3)$|SSIM|PSNR|AUC|Bal. Acc.|
> |-|-|-|-|-|
> |(10,100,10)|0.864|31.916|0.855|0.740|
> |(50,100,10)|0.846|30.112|0.833|0.731|
> |(10,500,10)|0.851|30.815|0.830|0.725|
> |(10,100,50)|0.861|31.006|0.851|0.741|
> |(50,500,10)|0.838|29.512|0.825|0.719|
> |(10,500,50)|0.845|30.436|0.829|0.723|
> |(50,100,50)|0.846|29.982|0.831|0.729|
>
> Over a 5X scaling of individual terms, SSIM ranged from 0.838–0.864, PSNR from 29.5–31.9, AUC from 0.825–0.855, and balanced accuracy from 0.719–0.741, i.e., **all metrics move by at most approximately 2-3% points in absolute value (and $\leq$ approximately 8% relative for PSNR).**

---

### Official Review · Reviewer_vRbn · 2025-11-06

**Soundness:** 2
**Presentation:** 2
**Contribution:** 1
**Rating:** 4
**Confidence:** 4

**Summary:**

The paper tackles the clinical MRI time–quality trade-off by formulating k-space acquisition under a fixed scan-time budget as a discrete RL problem and proposing E-DDPG, a DDPG-based agent that jointly optimizes sampling patterns, reconstruction quality, and downstream diagnostic accuracy. The method introduces three main components: a composite reward that balances cross-entropy reduction, SSIM improvement, and lower predictive entropy; a percentile-based replay buffer that balances learning from low- and high-value transitions; and a straight-through Gumbel-Softmax scheme that permits discrete top-K action selection while keeping gradients end-to-end. Experiments on fastMRI/fastMRI+ knee data show superior performance over prior RL baselines.

**Strengths:**

1.The paper tackles a problem with clear clinical value by jointly optimizing sampling patterns, reconstruction quality, and downstream diagnostic accuracy.
2. This paper adapts DDPG to MRI acquisition with a clean one-step formulation (“directly select K lines”), uses Straight-Through Gumbel-Softmax to handle the non-differentiable top-K action, and employs a percentile-based replay strategy to stabilize off-policy learning.
3. On fastMRI/fastMRI+ knee data, the approach improves both image quality metrics and downstream classification compared to strong RL baselines, supporting the practical promise of task-aware sampling.

**Weaknesses:**

1. The paper mainly ports a standard DDPG pipeline to MRI acquisition and combines it with known ingredients. This combination is more engineering-oriented and lacks practical innovation.
2. The reconstructor and classifier are frozen while the sampling policy changes. Different masks can interact with a fixed reconstructor’s generalization in unpredictable ways, so the reported quality may not reflect the true performance under the new sampling distribution.
3. Choosing all K lines in a single step (non-sequential) foregoes the possibility of adapting later choices to earlier observations, which is central to active/greedy sampling and closer to how scanners could operate.
4. Experiments focus on a single anatomy and task (knee, binary classification). The method’s utility should be validated on additional anatomies and tasks (multi-class, detection/segmentation), as well as across datasets.
5. The action space is tailored to 1D phase-encode top-K selection. It does not naturally extend to 2D sampling patterns (e.g., random, variable-density, radial/spiral) or trajectory constraints. At minimum, the paper should discuss this limitation or provide preliminary extensions.

**Questions:**

1. Could you please either add a small sequential variant (choosing k lines per step) or at least provide a preliminary conceptual modification of your current framework to support sequential sampling?
2. Since the reconstructor and classifier are fixed, could a mismatch between the learned mask and the frozen models bias the results? Please clarify how you avoid this issue, or provide (i) results with retraining or brief fine-tuning under the learned mask, and (ii) robustness checks across different reconstructors and classifiers (e.g., UNet, ViT).
3. The action space targets 1D top-K columns. How does the approach extend to 2D Cartesian patterns or non-Cartesian trajectories (random, variable-density, radial/spiral) and scanner constraints? If not supported, please discuss this limitation.
4. Can you add at least one more anatomy/task or dataset and strengthen statistical evidence (e.g., p-values)?
5. How are ACS/central lines enforced in the action space, and how sensitive are results to ACS size? A brief clarification would help.

---

> ### Author Response · Authors · 2025-12-02
> **Part-1/3**
>
> > **R1.1 & R3.2.** Could you please either add a small sequential variant (choosing k lines per step) or at least provide a preliminary conceptual modification of your current framework to support sequential sampling?
>
> We add a sequential (or active sampling) variant of the current framework, where the agent selects a small set of columns at each step until the budget is exhausted. Two scenarios were considered: 1) 4-column MDP (i.e., 4 columns per step) and 2) 10-column MDP (i.e., 10 columns per step). The results are shown below together with those from the 1-step MDP.
>
> |Acceleration|Method|SSIM|PSNR|AUC|Bal. Acc.|
> |-|-|-|-|-|-:|
> |4X|4-column|0.814|29.91|0.784|0.695|
> ||10-column|0.844|30.91|0.845|0.734|
> ||1-step MDP|0.864|31.91|0.855|0.740|
> |8X|4-column|0.800|28.32|0.760|0.675|
> ||10-column|0.840|29.66|0.830|0.725|
> ||1-step MDP|0.863|31.10|0.847|0.744|
> |10X|4-column|0.781|27.62|0.725|0.657|
> ||10-column|0.831|29.02|0.811|0.702|
> ||1-step MDP|0.853|30.84|0.831|0.717|
>
> **Across all accelerations, performance increased when more columns were selected per step.** This behavior likely originates from the current reward design, Gumbel-Softmax policy, and hyperparameters tuned for a 1-step MDP, rather than any inherent weaknesses of sequential MDPs. A systematic and in-depth analysis is required to draw a fair conclusion, most importantly including latency introduced by active sampling due to real-time constraints of the MRI pulse-generation hardware (**please see R3.2 for details**).
>
> > **R1.3.** The action space targets 1D top-K columns. How does the approach extend to 2D Cartesian patterns or non-Cartesian trajectories (random, variable-density, radial/spiral) and scanner constraints? If not supported, please discuss this limitation.
>
> Action spaces for 2D Cartesian patterns, stacks of 2D non-Cartesian radial/spiral readouts, and fully 3D non-Cartesian trajectories (e.g., 3DPR, Cones, FLORET, Yarnball) can all be mapped to 1D top-K indices. In 3D Cartesian imaging (i.e., 2D Cartesian patterns), the phase-slice encoding matrix can be vectorized to a 1D index set. For 2D and 3D non-Cartesian imaging, gradient waveforms are typically precomputed as a small set of representative templates, and the full trajectory is generated by applying rotation matrices to achieve the prescribed FOV and resolution. **This process naturally produces an ordered list of "spokes/interleaves", which is a 1D view-order (index) that is compatible with a 1D action space. Both Cartesian and non-Cartesian trajectories therefore admit a 1D view-order representation equivalent to the column index used in our work.** The top-K operator then selects the specific spokes/interleaves or phase-slice encoding lines to acquire. Although such extensions into existing or prospectively acquired labeled datasets (publicly unavailable) is outside the present scope, we have included this in the Discussion.
>
> > **R1.4 & R2.2.** Can you add at least one more anatomy/task or dataset and strengthen statistical evidence (e.g., p-values)?
>
> As suggested, we add an additional task targeting sprain-injury detection (training, validation and test dataset consist of 5.00\%, 2.08\%, and 5.59\% positive labels) using the same acquisition and evaluation pipeline. **Please see our detailed response to R2.2 regarding sprain-injury detection.**
>
> We also strengthen the statistical evidence by reporting p-values from paired t-tests computed across four independent runs when comparing E-DDPG against the baselines. **At a 0.05 significance level, the proposed method differs significantly from the comparing methods for most metrics.** p-values are shown below, with values > 0.05 in boldface:
>
> |Acceleation|Metric|ASMR|Pineda|Xu & Oksuz|
> |-|-|-|-:|-:|
> |4X|SSIM|0.0357|<1e-4|0.0003|
> ||PSNR|**0.0511**|0.0016|0.0222|
> ||AUC|0.0494|0.0157|0.0239|
> ||Bal. Acc.|0.0195|0.0012|0.0375|
> |8X|SSIM|0.0024|<1e-4|0.0010|
> ||PSNR|0.0357|<1e-4|**0.0522**|
> ||AUC|0.0122|<1e-4|0.0402|
> ||Bal. Acc.|0.0189|1e-4|0.0425|
> |10X|SSIM|0.0020|<1e-4|0.0024|
> ||PSNR|0.0046|0.0007|0.0037|
> ||AUC|0.0307|0.0035|0.0029|
> ||Bal. Acc.|0.0011|<1e-4|0.0122|

---

> ### Author Response · Authors · 2025-12-02
> **Part-2/3**
>
> > **R1.2 & R2.1** Since the reconstructor and classifier are fixed, could a mismatch between the learned mask and the frozen models bias the results? Please clarify how you avoid this issue, or provide (i) results with retraining or brief fine-tuning under the learned mask, and (ii) robustness checks across different reconstructors and classifiers (e.g., UNet, ViT).
>
> The reviewer raises a valid point—such a mismatch could indeed introduce bias. The scope of this work, however, centers on the RL algorithm, which operates on a fixed reconstruction backbone (PromptMR) and a fixed classifier (ResNet-50). Any bias arising from this mismatch affects all comparison methods to a similar degree; thus, the core contribution and performance trends remain intact. Nevertheless, we provide the requested fine-tuning experiment under each method’s learned mask. We begin with the ResNet-50 classifier and replace its final fully connected layer with a three-layer MLP head. For each method and acceleration factor, we freeze the entire ResNet-50 backbone and fine-tune only the three-layer MLP head. This head is initialized from the classifier trained on fully sampled reconstructions and then optimized for 10 epochs with lr = 1e-5, using reconstructions generated by each method’s learned sampling pattern at the corresponding acceleration. We subsequently evaluate each policy using its fine-tuned classifier and reconstructor. The results are shown below, with boldface indicating values that surpass those of the proposed E-DDPG.
>
> |Acceleration|Method|SSIM|PSNR|AUC|Bal. Acc.|
> |-|-|-:|-:|-:|-:|
> |4X|E-DDPG|0.864|31.916|0.864|0.751|
> ||ASMR|0.854|**32.250**|0.853|0.749|
> ||Pineda et al.|0.816|28.124|0.811|0.689|
> ||Xu & Oksuz|0.851|31.271|0.843|0.744|
> |8X|E-DDPG|0.863|31.106|0.861|0.753|
> ||ASMR|0.850|30.453|0.844|0.741|
> ||Pineda et al.|0.791|27.682|0.789|0.682|
> ||Xu & Oksuz|0.840|30.320|0.831|0.738|
> |10X|E-DDPG|0.853|30.840|0.833|0.726|
> ||ASMR|0.844|29.198|0.825|0.722|
> ||Pineda et al.|0.782|27.212|0.741|0.676|
> ||Xu & Oksuz|0.836|30.121|0.824|0.723|
>
> Other than ASMR's PSNR at 4X, all remaining metrics indicate that E-DDPG outperforms the comparing methods across all acceleration factors.
>
> Regarding robustness checks, we used a U-Net–based architecture for both the image reconstructor and classifier across all evaluated algorithms. **Please see R2.1 for results using the U-Net reconstructor only.** The classifier follows a standard U-Net design: the encoder processes the input through a sequence of blocks, each comprising two Conv→BN→ReLU layers, and the decoder restores spatial resolution via transposed convolutions. The reconstructor is adapted from the official fastMRI U-Net implementation (https://github.com/facebookresearch/fastMRI/tree/main/fastmri_examples/unet). All models were trained for acceleration factors of 4X, 8X, and 10X using these updated classifier and reconstructor backbones, and the results are shown below.
>
> |Acceleration|Method|SSIM|PSNR|AUC|Bal. ACC.|
> |-|-|-:|-:|-:|-:|
> |4X|Baseline|1.000|Inf|0.780|0.664|
> ||E-DDPG|0.830|29.112|0.762|0.630|
> ||ASMR|0.821|28.980|0.753|0.622|
> ||Pineda et al.|0.787|26.598|0.726|0.588|
> ||Xu & Oksuz|0.818|28.516|0.749|0.618|
> |8X|Baseline|1.000|Inf|0.780|0.664|
> ||E-DDPG|0.825|28.991|0.757|0.619|
> ||ASMR|0.806|28.215|0.736|0.601|
> ||Pineda et al.|0.754|25.512|0.702|0.563|
> ||Xu & Oksuz|0.804|28.116|0.732|0.606|
> |10X|Baseline|1.000|Inf|0.780|0.664|
> ||E-DDPG|0.813|28.217|0.739|0.581|
> ||ASMR|0.801|27.987|0.718|0.574|
> ||Pineda et al.|0.744|23.965|0.689|0.526|
> ||Xu & Oksuz|0.796|27.891|0.710|0.569|
>
> Therefore, **together with the results in R2.1 we conclude that our algorithm remains robust across different reconstructors and classifiers.**

---

> ### Author Response · Authors · 2025-12-02
> **Part-3/3**
>
> > **R1.5.** How are ACS/central lines enforced in the action space, and how sensitive are results to ACS size? A brief clarification would help.
>
> As described in Section 2.2, the ACS/central lines are enforced through the definition of action space in Eq. (2). The predefined ACS lines are always included as $m_\text{init}$ and are not part of the learnable action space; the agent only selects from the remaining non-ACS lines. The same definition and implementation were applied to all other comparing algorithms.
>
> Regarding sensitivity, we conducted an additional experiment at 4X acceleration on the same dataset with different numbers of ACS lines—16 (5%), 22 (7%), and 32 (10%).
>
> |ACS|Method|SSIM|PSNR|AUC|Bal. Acc.|
> |:-|:-|-:|-:|-:|-:|
> |16 (5%)|E-DDPG|0.864|31.916|0.855|0.740|
> ||ASMR|0.854|32.250|0.838|0.737|
> ||Pineda et al.|0.816|28.124|0.801|0.677|
> ||Xu & Oksuz|0.851|31.271|0.835|0.735|
> |22 (7%)|E-DDPG|0.869|32.585|0.863|0.748|
> ||ASMR|0.862|32.671|0.859|0.750|
> ||Pineda et al.|0.831|29.512|0.811|0.699|
> ||Xu & Oksuz|0.859|31.912|0.855|0.746|
> |32 (10%)|E-DDPG|0.875|33.240|0.870|0.756|
> ||ASMR|0.869|33.106|0.866|0.754|
> ||Pineda et al.|0.834|30.671|0.816|0.711|
> ||Xu & Oksuz|0.864|32.614|0.859|0.750|
>
> As shown above, **the performance of E-DDPG in reference to the comparison methods remains largely unchanged.**
>
> > **R1.6** The reconstructor and classifier are frozen while the sampling policy changes. Different masks can interact with a fixed reconstructor’s generalization in unpredictable ways, so the reported quality may not reflect the true performance under the new sampling distribution.
>
> The reviewer is correct and raises a valid point. We note, however, that all compared methods were implemented under the same protocol: both the reconstructor and classifier were kept frozen so that the evaluation focuses on the effect of the RL sampling policy. Any mismatch between a learned sampling distribution and the fixed backbone therefore applies equally to all algorithms, including E-DDPG. We agree that jointly retraining the reconstructor and classifier for each policy would represent a different, more end-to-end regime. This broader direction is already discussed in the limitations section. In the revision, **we will make explicit that the top-performing baselines (Xu & ASMR) also use the same frozen backbone, to make sure that our comparisons are controlled and focus solely on the sampling policy.**

---

### Author Response · Authors · 2025-12-02
**Summary of the Response**

We thank the reviewers and ACs for their constructive suggestions and their time and effort amid the recent OpenReivew incident. We use this comment block to assist the ACs by providing a clear sense of the overall consensus across the reviews and by outlining our responses to address the reviewers' concerns.

All reviewers are unanimous in recognizing the following **strengths** of the paper:

1. The paper addresses a clinically important and well-motivated problems in accelerated MRI.
2. The formulation of $k$-space sampling as a discrete RL problem with a composite reward, percentile replay, and Straight-Through Gumbel-Softmax (STGS) is technically sound, coherent, and compleling.
3. The empirical results are consistent, with clear improvements over existing RL baselines and transparent ablation studies.

Throughout the rebuttal/revision period, we kept these strengths.

The reviewers express several **mixed views** and raise **questions**:

1. The assessment of novelty varies: **R1** considers the work primarily engineering-oriented, whereas **R3** views the engineering design as a meaningful and creative contribution.
2. Additional experiments on sequential versus 1-step decision-making were suggested (**R1**) and closely-related questions were raised (**R3**).
3. Concerns about dependence on the PromptMR reconstructor, reward-sensitivity analysis, and missing comparisons to differentiable sampling methods were raised by **R2** and **R3**.
4. **All reviewers** note the limitation to the current experiment on knee/meniscus tear, suggesting another pathology or anatomy; however, they differ on how strongly this should weight against the contributions.

We performed suggested experiments in greater depth to address all of the reviewers' questions and concerns. The main updates are summarized below in the order they appear in our response.

1. Active sampling experiments were conducted under two alternative acquisition strategies (**R1.1 & R3.2**).

2. Clarification and added discussion on the extensibility of the proposed method to 3D Cartesian and non-Cartesian MRI (**R1.3**).

3. An additional experiment was performed on a different pathology to assess generalization (**R1.4 & R2.2**).

4. A U-Net-based reconstructor was integrated into all algorithms and comparative analyses were performed (**R1.2 & R2.1**).

5. Method-specific classifier fine-tuning was performed for all algorithms under their own learned sampling patterns (**R1.2 & R2.1**).

6. A U-Net–based reconstructor and classifier pair was incorporated into all algorithms and comparative analyses were performed (**R1.2 & R2.1**).

7. The impact of ACS size was examined through a dedicated sensitivity analysis (**R1.5**).

8. Clarification regarding concerns that the reported quality may not reflect the method’s true performance (**R1.6**).

9. The effect of varying the reward coefficients was systematically analyzed (**R2.3**).

10. Two new algorithms were included in the comparative analysis (**R3.1**).

Finally, regarding the mixed view on novelty, we emphasize that our contribution fits within the **Healthcare & Applications to Physical Sciences** listed in the ICLR 2026 Subject Areas.

---

### Meta-Review · Area_Chair_KCJR · 2025-12-27

**Summary:**

The reviewers agree that the paper targets an important clinical problem (accelerated MRI acquisition under a fixed sampling budget) and presents an RL formulation to address it. However, the overall recommendation trends negative. The core technical contribution is largely viewed as an applied, engineering-driven integration, with limited evidence that the proposed approach generalizes beyond the specific experimental setting considered.

Based on the reviews, the AC believes that several key design choices are also insufficiently motivated (eg., the use of DDPG for a largely single-step). The rebuttal further introduces ambiguity in this regard: while a sequential formulation (arguably more principled for acquisition) is explored, its inferior performance is attributed to limited tuning rather than offering a clear justification. In addition, reviewers consistently raised concerns about the narrow scope of the empirical evaluation, in particular, the focus on a single anatomy and a binary diagnostic task. While an additional experiment was added, these concerns were not fully alleviated. Taken together, these issues lead the AC to recommend ``rejection`` sadly.

**Reviewer Concerns:**

The rebuttal addressed several requests from the reviewers. In particular, the authors explored sequential multi-step acquisition, evaluated robustness to the choice of reconstruction backbone by including U-Net–based models, and provided additional analyses on robustness to hyperparameters and ACS size. However, two fundamental concerns remain unresolved. Reviewers **vRbn** and **4ZVa** raised persistent questions regarding the conceptual novelty of the approach, as the contribution largely reflects an engineering integration rather than a substantive methodological advance. In addition, all reviewers expressed concern about the limited scope of the empirical evaluation: although an additional experiment was included in the rebuttal, the study remains largely confined to binary problems within a narrow experimental setting.

**Reviewer Scores:**

Given the rebuttal additions, the AC expects reviewers would acknowledge that a few of their concrete requests were addressed. That said, the AC does **not** expect any reviewer to champion acceptance because the concerns are mostly about *the nature of the contribution* (application novelty, limited clinical/setting breadth, and protocol dependence). So scores likely remain unchanged. The AC reads here:

Reviewer **vRbn (Rating 4, Confidence 4):** and Reviewer **4ZVa (Rating 4, Confidence 3)** likely **stays at 4**. Reviewer **hnqi (Rating 6, Confidence 3)**  likely stays at 6 (or even reduce to 4) if there was a chance to discuss the paper further.

---

### Decision · Program_Chairs · 2026-01-26

Reject